# Evaluating [^18^F]FDG and [^18^F]FLT Radiotracers as Biomarkers of Response for Combined Therapy Outcome in Triple-Negative and Estrogen-Receptor-Positive Breast Cancer Models

**DOI:** 10.3390/ijms241814124

**Published:** 2023-09-15

**Authors:** Paolo Rainone, Silvia Valtorta, Chiara Villa, Sergio Todde, Massimiliano Cadamuro, Gloria Bertoli, Donatella Conconi, Marialuisa Lavitrano, Rosa Maria Moresco

**Affiliations:** 1Department of Medicine and Surgery, University of Milano—Bicocca, 20900 Monza, Italy; paolo.rainone@unimib.it (P.R.); chiara.villa@unimib.it (C.V.); sergio.todde@unimib.it (S.T.); donatella.conconi@unimib.it (D.C.); marialuisa.lavitrano@unimib.it (M.L.); 2Nuclear Medicine Department, IRCCS San Raffaele Scientific Institute, 20132 Milan, Italy; silvia.valtorta@ibfm.cnr.it; 3Institute of Molecular Bioimaging and Physiology, National Research Council (IBFM-CNR), 20054 Segrate, Italy; gloria.bertoli@ibfm.cnr.it; 4NBFC National Biodiversity Future Center, 90133 Palermo, Italy; 5Tecnomed Foundation, University of Milano—Bicocca, 20126 Monza, Italy; 6Department of Medicine (DIMED), University of Padua, 35128 Padua, Italy; massimiliano.cadamuro@unipd.it; 7General Internal Medicine Unit, Padua University-Hospital, 35128 Padua, Italy

**Keywords:** breast cancer, targeted therapy, prognostic biomarkers, PET/CT, metformin, syrosingopine

## Abstract

Breast cancer (BC) is the most frequent cancer and the second leading cause of death in women. A typical feature of BC cells is the metabolic shift toward increased glycolysis, which has become an interesting therapeutic target for metabolic drugs such as metformin (MET). Recently, the administration of the antihypertensive syrosingopine (SYRO) in combination with MET has shown a synergistic effect toward a variety of cancers. However, a fundamental need remains, which is the development of in vivo biomarkers that are able to detect early clinical response. In this study, we exploited a triple-negative murine BC cell line (4T1) and a metastatic ER+ murine BC cell line (TS/A) in order to investigate, in vivo, the early response to treatment, based on MET and/or SYRO administration, evaluating [^18^F]FDG and [^18^F]FLT as potential biomarkers via PET/CT. The study provides evidence that SYRO plus MET has a synergistic effect on tumor growth inhibition in both 4T1 and TS/A experimental models and has showed the highest efficacy on the TNBC xenograft mice (4T1) via the expression reduction in the lactate transporter MCT4 and in the epithelial–mesenchymal transition biomarker Snail, promoting its potential application in therapy settings. In addition, the selective reduction in the [^18^F]FLT tumor uptake (at 7 dd), observed in the SYRO plus MET treated mice in comparison with the vehicle group, suggests that this radiotracer could be potentially used as a biomarker for the early detection of therapy response, in both evaluated xenografts models.

## 1. Introduction

Breast cancer (BC) is the most common cancer among women, according to the recent report GLOBOCAN 2020 [1] of the International Agency for Research on Cancer; the new cases of BC in 2020 were 2.26 million. Despite recent advances in diagnosis and treatment having led to a reduction in incidence and mortality, it remains the second leading cause of cancer-related death in women. Among different subtypes of BC, triple-negative phenotype (TNBC) accounts for approximately 15–20% of all invasive BCs. The high aggression and lack of specific therapeutic targets limit the treatment options and make its prognosis dismal [2]. Cisplatin, anthracyclines, and taxanes administrated alone or in combination are the standard chemotherapy for patients with TNBC, during neoadjuvant treatment. Unfortunately, patients who have not achieved pathologic complete response have a high relapse rate. Some studies based on the use of immuno-check point or cyclin-dependent kinase 4/6 inhibitors are ongoing, but the majority of reported results are at the preclinical stage or are based on early-phase clinical trials [3]. 

A typical feature of cancer cells including BC is the capability of metabolic reprogramming with a shift toward an increased glycolysis. For this reason, in recent years, the interest in drug targeting for tumor metabolism, such as the antidiabetic drug metformin (MET), was raised [4]. Although pieces of clinical evidence are not yet available, recent studies showed that MET was able to reduce resistance to cisplatin in preclinical models of cancer including TNBC [5]. The mechanism of action involves a blocking of the mitochondrial respiratory chain, leading to the reduction in oxidative phosphorylation (OXPHOS) activity and the depletion of ATP production. This effect induces an increment of the AMPK-mediated catabolic versus the anabolic processes, with a consequent reduction in cell proliferation and apoptosis activation [6,7]. Recently, Lord et al. [8] described a bimodal molecular response to MET in patients with BC, with the first consisting in a reduction in the OXPHOS transcriptional pathway (MET sensitive) with an increased expression of multiple genes regulating glycolysis and glucose transport, and the second is the OXPHOS transcriptional response (OTR) group for which there is an increase in OXPHOS gene transcription with an increment of tumor proliferation. Interestingly, patients in whom OXPHOS activity was inhibited displayed higher uptake levels of the glucose analog [^18^F]FDG at PET scans during MET treatment compared to those with lower OXPHOS sensitivity (MET resistant). Intriguingly, all patients in this last group (OTR) carried an estrogen receptor positive (ER+) tumor phenotype. Therefore, [^18^F]FDG uptake and/or ER expression might also act as biomarkers to distinguish the two types of metabolic response. On the other hand, several studies have indicated that the cancer cells subtype with a preference for aerobic glycolysis developed resistance to OXPHOS-targeting inhibitors, including MET [9,10]. For these reasons, combinatory treatment able to counteract the glycolytic response to MET might improve tumor sensitivity [11]. A recent work assessed that monocarboxylate transporter 4 (MCT4), a lactate transporter, was highly expressed in the aerobic glycolysis-preference subtype with functions supporting the proliferation of these cells [12]. Inhibitors of mono-carboxylate transporters represent a target of interest to block the extrusion of the increased lactate levels in a tumor microenvironment associated with the hyper-activation of glycolytic pathway. Indeed, the toxic intracellular acidification secondary to the increased lactate production during aerobic glycolysis is mitigated by the excretion of lactate and H^+^ ions via MCT4 [13]. On the other hand, monocarboxylate transporter 1 (MCT1) allows taking the excess of lactate by the cells with an oxidative phenotype [14]. MCT1 exhibits widespread expression throughout the body [15], while MCT4 is predominantly found in highly glycolytic tissues [16,17,18], and their silencing has been shown to reduce tumor growth in different xenografts models of cancer [19,20,21,22,23]. Therefore, drugs able to target the lactate transporters MCT1 and MCT4 represent a potential strategy to overcome increased glycolysis and energetic symbiosis between cancer cells. The antihypertensive drug syrosingopine (SYRO) is a dual inhibitor of MCT1 and MCT4, which increases intracellular lactate levels [24], inducing a ROS (reactive oxygen species)-dependent cellular apoptosis [12]. In combination with MET, SYRO provides a synergistic lethal effect toward a variety of cancer cell lines in vitro and in vivo on hepatocarcinoma but not against untransformed cells, thus representing a novel option in cancer therapy [25,26]. In light of the effect of metabolic drugs on glycolysis and cell proliferation, the use of [^18^F]FDG and [^18^F]FLT that traces the first step of glycolysis and cell proliferation, respectively, represents potential biomarkers of response. However, both radiopharmaceuticals have been evaluated in preclinical [27] and recently in clinical [28] settings of BC, for the in vivo evaluation of response to standard chemotherapy including paclitaxel and aromatase inhibitors. 

In the present study, we investigated the therapeutic efficacy of the SYRO and MET combination in comparison with the single treatments or cisplatin administration in two different BC xenograft mouse models. To clarify the molecular effects and to search for potential biomarkers of response, drug efficacy was monitored in vivo via PET imaging using different radioligands and post mortem via molecular analysis.

## 2. Results

### 2.1. Evaluation of Treatment Efficacy

For the 4T1 xenograft model, two sets of experiments were performed because the starting study was stopped after six days of treatment for problems linked to the COVID-19 pandemic, and it was considered as a preliminary efficacy study. In this first experimental session, the effect of DMSO (as a vehicle of SYRO drug solution) was also evaluated, which has not reported notable effects. The preliminary data, six days post treatment, showed that only SYRO plus MET administration significantly reduced (*p* < 0.05) tumor volume (Appendix A) with a tumor growth inhibition (%TGI) of 51.0% (Appendix A), confirming a synergistic effect for the latter therapy. For the other treatment groups, the results revealed a %TGI of 19.3, 29.1, 33.4, and 23.6% for cisplatin, MET, cisplatin plus MET, and SYRO, respectively. In the second session, the study was replicated, and similar efficacy results were observed after ten days of treatment. Data confirmed a synergistic effect for SYRO plus MET treatment, reaching a %TGI over 60% (Figure 1a) even after seven days (65.66% and 63.25% at 7dd and 10 dd, respectively); in fact, it was able to remarkably reduce the tumor size (*p* < 0.01) (Figure 1b). Among the other treatments, cisplatin showed a lower but significant reduction in tumor volume (*p* < 0.05) with a %TGI of 38.6%. Lower efficacy was revealed for the MET (10.9%), MET plus cisplatin (21.5%), and SYRO (26.9%). The efficacy and synergy of SYRO plus MET administration was also confirmed in a TS/A xenograft mouse model, with a %TGI of about 40% (40.55%) (Figure 2a) and a significant reduction in tumor volume (*p* < 0.05) (Figure 2b) after eleven days of treatment. The cisplatin treatment group shows a considerable but not significant trend of tumor volume reduction and a %TGI of 26.8%. A slighter effect was shown with the MET (9.40%), MET plus cisplatin (9.53%), and SYRO groups (2.53%).

### 2.2. PET/CT Studies

For the 4T1 xenograft mouse model, PET images performed at seven days post treatment (Figure 3a) showed a significant reduction in [^18^F]FLT tumor uptake in cisplatin plus MET (*p* < 0.05) and in SYRO plus MET (*p* < 0.01) groups (Appendix A), compared to the acquisitions performed on animals before the start of treatment, whereas [^18^F]FDG uptake unspecifically increased in all experimental groups (Appendix A), independently to therapy efficacy. Notably, at the end of treatment, only the SYRO plus MET group showed a reduction in [^18^F]FLT tumor uptake in comparison with control animals (*p* < 0.05) (Figure 4b), whilst [^18^F]FDG uptake remained stable among groups (Figure 4a). Differently to what was observed in 4T1, in TS/A cells implanted mice (Figure 3b), PET results (at 7 dd) revealed a significant reduction in [^18^F]FDG tumor uptake in cisplatin plus MET (*p* < 0.05), SYRO (*p* < 0.05), and SYRO plus MET (*p* < 0.01) groups, in comparison with baseline condition (Appendix A), whereas [^18^F]FLT uptake remained stable over time in all groups (Appendix A). However, at seven days, a significant reduction in [^18^F]FLT tumor uptake was observed only in SYRO-plus-MET-treated mice in comparison with the control group (*p* < 0.05) (Figure 5b). Conversely, a decrease in [^18^F]FDG tumor uptake was reported in all groups, regardless of the effects on tumor growth (Figure 5a).

### 2.3. Ex Vivo Molecular Assay

Molecular analyses via RT-qPCR demonstrated a significant decrease in the lactate/H^+^ transporter *MCT4* levels in the tumor tissue of the 4T1 xenograft mouse model treated with both cisplatin and MET plus SYRO (*p* < 0.05) (Figure 6b), whereas low levels of the epithelial–mesenchymal transition (EMT) biomarker *Snail* (Figure 6e) were observed only in the group treated with the combination of MET plus SYRO (*p* < 0.05). IHC analyses confirmed a significant decrease in MCT4 levels in the 4T1 xenograft mouse model treated with both cisplatin and MET plus SYRO (*p* < 0.01) (Figure 7), and remarkable differences were not reported for other analyzed biomarkers (Appendix A). On the TS/A xenograft mouse model, significant variations among the evaluated biomarkers were not reported via RT-qPCR (Appendix A).

## 3. Discussion

In the present study, we first evaluated the efficacy of SYRO plus MET on tumor growth, compared to different treatments, in TNBC (4T1) and metastatic ER+ (TS/A) xenograft mouse models. On 4T1 mice, data revealed a synergistic effect on tumor growth inhibition for SYRO plus MET administration starting at an early time (at 6/7 days post treatment). The effect showed using the SYRO and MET combination was even higher than cisplatin administration, used as standard therapy for BC patients. No significant effect was reported with the other tested therapeutic strategies. In the TS/A xenograft model, only the SYRO plus MET group showed a significant reduction in tumor growth with a comparable efficacy to what was observed in 4T1 mice at the end of treatment (at 11 days). Thus, these data indicate that the SYRO plus MET combination is able to reduce tumor growth in both syngeneic models of BC, with a higher effect than cisplatin treatment. Neither single MET administration nor the combination with cisplatin was able to reduce the volume increase of the lesions. 

To investigate the molecular mechanism and to search for potential in vivo biomarkers of therapy response, the tumor uptake of [^18^F]FLT (cell proliferation) and [^18^F]FDG (glucose metabolism) radiotracers was measured via PET/CT, before and during treatment administration (at 7 days). In both 4T1 and TS/A models, we observed, in PET studies, a significant reduction in tumor uptake using the proliferation marker [^18^F]FLT, exclusively in the SYRO plus MET group. On the contrary, [^18^F]FDG uptake was differently modulated in these two models, with no significant variations reported in the 4T1 tumor and an independent effect to therapy response in the TS/A mice. Overall, the selective reduction in the [^18^F]FLT tumor uptake, observed in the SYRO-plus-MET-treated mice in comparison with the vehicle group (at 7 days post treatment), suggests that this radiotracer could potentially serve as a biomarker for the detection of therapy response, even early that tumor volume reduction occurred, in both xenografts models. Meanwhile, results obtained with [^18^F]FDG indicate the limitations of this radiotracer in evaluating treatment outcomes for such a kind of combined therapy strategy. In order to explore the mechanisms underlying the synergistic effect of SYRO plus MET, and to better understand the molecular meaning of [^18^F]FLT and [^18^F]FDG tumor uptake modulation, an ex vivo molecular analysis on tumor tissue was performed. In particular, biomarkers of proliferation, differentiation, proton/lactate transport, and cell invasiveness were included in the assay. The RT-qPCR results on the TNBC (4T1) samples indicated a significant decrease in MCT4 levels (not for MCT1) after treatment with SYRO plus MET, as also confirmed with the IHC assay, whereas this effect was not shown in the TS/A model. As previously stated [12], the knockdown of MCT4 in non-small cell lung cancer increased intracellular lactate levels inducing ROS (reactive oxygen species)-dependent cellular apoptosis in the aerobic glycolysis preference cell subtype. Likewise, our findings on TNBC (4T1) tumor tissue confirmed that the synergistic efficacy of SYRO plus MET administration is associated with the inhibition of MCT4 expression, likely mediated using SYRO. In addition, a significant reduction in the epithelial–mesenchymal transition (EMT) biomarker *Snail* was exclusively observed in 4T1 tumor tissue treated with the SYRO and MET combination, also confirming its efficacy in hampering this pathway [29,30] associated with a poor prognosis in BC patients [31]. Indeed, the lactate-inducible Snail protein is involved in the regulation of glucose flux toward the pentose phosphate pathway (PPP), allowing cancer cell survival under metabolic stress [32]. For this reason, the lower transcription levels of *Snail* observed in combination with MCT4 reduction may additionally increase the vulnerability of 4T1 tumor lesions to the metabolic stress promoted by the SYRO and MET combination. However, a synergistic effect was also reported in the TS/A model, where both transporters (MCT4 and MCT1) are minimally expressed and not modulated by treatments. This observation suggests that other molecular mechanisms should be involved in the combined effect of SYRO and MET on TS/A mice, such as an involvement of ER modulation. In this regard, the use of [^18^F]Fluoroestradiol, a widely employed tracer for the in vivo detection of estrogen receptor density [33,34], could be of interest for a better understanding of the ER levels’ involvement in drugs that modulate cellular metabolism.

Our findings on [^18^F]FDG tumor uptake revealed a different sensitivity of 4T1 and TS/A cells to drug-induced metabolic stress, with no significant modification in the first and a clear reduction in the latter. Benjamin et al. [25] showed in 6.5R0 cells, a line exclusively dependent via glycolysis for ATP generation, a dramatic reduction in both ATP and lactate production post SYRO treatment. Accordingly, we could speculate that a double blocking of energy production, i.e., upstream glycolysis by SYRO plus OXPHOS by MET, is associated with the effect observed in TS/A model. Finally, despite the reduction in [^18^F]FLT tumor uptake observed in both models, we fail to find any treatment-related modifications in Ki67 expression. [^18^F]FLT is trapped into the cells via thymidine kinase 1, an enzyme hyper-expressed in the S phase of cell cycle [35]. While Ki67 levels are upregulated during the whole cycle phase, but not in G0 and for this reason, it is used as proliferation marker. Our data suggest that the SYRO plus MET therapy should mainly affect cancer cells in G1/S phase, prior to entrance in S phase of the cell cycle [36]. 

In conclusion, our in vivo study results confirm SYRO plus MET as a promising treatment for cancer in general and particularly for BC. However, molecular and in vivo imaging results clearly indicate that a simple combined blocking of OXPHOS and proton extrusion pumps (MCT4) is not sufficient to fully explain the efficacy of this combined treatment, and further investigations are required to better clarify the mechanisms underlying the synergistic effect of SYRO plus MET in these BC models.

## 4. Materials and Methods

### 4.1. Cell Culture

The 4T1 and TS/A spontaneous murine cells are TNBC and ER+ phenotypic lines, respectively, originated in BALB/C mice [37,38,39,40]. The 4T1 (ATCC) cells were cultured in RPMI 1640 medium supplemented with 10% FBS, 100 units of penicillin, and 100 μg/mL of streptomycin. TS/A cells (Sigma-Aldrich Inc., St. Louis, MO, USA) were grown in Dulbecco’s modified MEM (GIBCO, Life Technologies, Monza, MB, Italy) supplemented with 2 mM of glutamine, 100 U/mL of penicillin, 100 μg/mL of streptomycin, and 10% heat-inactivated fetal calf serum (GIBCO, Life Technologies, Monza, MB, Italy) at 37 °C in humidified 5% of CO_2_ atmosphere. Culture medium was changed every 2–3 days, and cells were passaged with 0.25% trypsin/EDTA. Cells were routinely tested for Mycoplasma using a MycoAlert mycoplasma detection kit (BioWhittaker-Lonza, Euroclone S.p.a., Milan, Italy).

### 4.2. Animal Model

Female Balb/c mice were obtained from the Charles River Laboratories Italia S.r.l. Animal experiments were carried out in compliance with the institutional guidelines for the care and use of experimental animals (IACUC) and national rules for experimental use of animals. The protocol has been approved by the Italian Ministry (approval number: 793/2020-PR). Balb/c nude mice, with the age of 7–8 weeks and weight of 24–26 g, were maintained under specific pathogen free conditions. To obtain tumor xenograft models, 5 × 10^4^ of either 4T1 or TS/A cells were subcutaneously injected in mice on the right flank, in 50 µL of PBS (phosphate-buffered saline) solution. Mice weight and tumor growth (digital caliper) were recorded twice a week until the end of the study.

### 4.3. In Vivo Efficacy Study

When the tumors were palpable, the animals inoculated with 4T1 or TS/A were randomly assigned to six groups of treatment: vehicle, cisplatin (3 mg/kg i.p, twice a week), MET (250 mg/kg i.p., daily), SYRO (7.5 mg/kg i.p., three times per week), cisplatin plus MET, and MET plus SYRO at the administration regimens indicated above. All drugs were dissolved in saline except SYRO that was dissolved in DMSO (10% in sterile water). Effects of DSMO on tumor growth were compared to saline solution. Because no effect was present, we used the last for comparison. Tumor volume was calculated following the formula: [length × (width)^2^]/2. The quantification of tumor growth variations was performed as (Tv_end_ − Tv_0_)/Tv_0_ × 100, where Tv_0_ represents the tumor volume at the beginning of the treatment (day 0), and Tv_end_ is the median tumor volume of the same mouse, determined on day 10 for 4T1 and day 11 for TS/A, to which corresponds the end of the treatment. Moreover, the tumor growth inhibition (%TGI) was evaluated using the following formula [41,42]: %TGI = [1 − (Tt/T0/Ct/C0)]/[1 − (C0/Ct)] × 100, and the impact of combined treatment was calculated at the several time points based on the methodology proposed by Navarro et al. [43] (52% and 39% were considered as thresholds for drug efficacy study on 4T1 and TS/A models, respectively). At the end of the experiment, mice were sacrificed under anesthesia, and the tumors were collected for postmortem molecular analyses.

### 4.4. PET/CT Images Analysis

PET imaging was performed with 3′-deoxy-3′-[^18^F]fluorothymidine ([^18^F]FLT) and 2-deoxy-2-[^18^F]fluoro-D-glucose ([^18^F]FDG) to assess proliferation related to TK1 expression and hexokinase activity, respectively. [^18^F]FLT and [^18^F]FDG uptake was evaluated via PET/CT imaging in distinct groups of mice at baseline and at 1 week after the beginning of treatment. PET/CT acquisitions were performed using β-cube^®^ and X-cube^®^ (Molecubes, Gent, Belgium), respectively. On the day of PET/CT imaging, animals were injected in a tail vein with 4.67 ± 0.3 MBq of either [^18^F]FDG or [^18^F]FLT radiotracer and maintained under a red warm lamp. After about 60 min of tracer uptake, each animal was anaesthetized with 2% isoflurane in medical air and then positioned prone on the scanner bed for the CT scan centered on the tumor (exam duration: 4 min, X-Ray beam duration: 90 s, kVp: 40, current: 400 μA, rotation time: 60 s, and angular views: 960). At the end of the CT acquisition, the bed with the immobilized animal was removed and inserted in the PET scanner for a 20 min static acquisition. During the exam, mice were maintained under anesthesia, and body temperature and respiratory rate were constantly monitored.

### 4.5. Total mRNA Extraction and Quantitative Real-Time PCR (RT-qPCR) Analysis

Total RNA was extracted and purified from breast tumor tissue using RNeasy Mini Kit (Qiagen, Hilden, Germany) and eluted in water. One microgram of the total extracted amount of RNA was subsequently treated with DNase I and reverse-transcribed using High-Capacity cDNA Reverse Transcription Kit (Thermo Fisher Scientific, Waltham, MA, USA). For a quantitative estimate of mRNA levels, a StepOne Real-Time PCR System (Thermo Fisher Scientific, USA) with dual-labeled TaqMan probes was used. For each target, assay-on-demand products were employed (assay IDs: Mm01306379_m1, Mm00446102_m1, Mm01278617_m1, Mm01333430_m1, Mm00441533_g1, Mm00441531_m1, Mm00442991_m1, Mm00439498_m1, and Mm00442991_m1 for Mct1, MCT4, Ki67, Vimentin, Snail, Slug, Mmp2, and Mmp9, respectively, Thermo Fisher Scientific, USA), and the relative amount of mRNAs were determined by comparison with the housekeeping Gapdh probe (Mm99999915_g1, Thermo Fisher Scientific, USA). Final volume reaction was 20 μL, using the TaqMan Gene Expression Master Mix (ABI 4369016, Thermo Fisher Scientific, USA). Cycle parameters used consisted of 2 min at 50 °C and 10 min at 95 °C, followed by 40 cycles of 15 s at 95 °C for denaturation and 1 min at 60 °C for annealing/extension. Inter-run calibrator was required to correct for possible run-to-run variation because all samples were not analyzed in the same run. Appropriate controls (no template and no enzyme) were added to each run. For the comparison, a calibrator specimen from the control group was analyzed on every assay plate with unknown specimens of interest. Relative mRNA levels were calculated as follows: 2 − [(DCt(sample) − DCt(calibrator)] = 2 − DDCt, where DCt equals Ct (molecule under analysis) − Ct (housekeeping gene). The data were presented as the fold change in gene expression normalized to an endogenous reference gene and relative to the control.

### 4.6. Immunohistochemistry (IHC) on Frozen Samples and Immunohistochemical Analysis

Briefly, acetone-fixed, 4 µm thick cut frozen tissue samples of tumor were immunostained with antibodies against MCT1 (Novus Biological, Centennial, CO, USA, 1:100), MCT4 (Novus Biologicals, 1:100), Vimentin (Novus Biologicals, 1:200), Ki67 (Novus Biological, 1:100), Snail (AbCAM, Cambridge, UK, 1:200), and Slug (AbCAM, 1:100). Briefly, following rehydration of the slides with phosphate-buffered saline (PBS) 1× (Euroclone S.p.a., Milan, Italy), unspecific binding was blocked, incubating samples for 10 min with UltraVision protein block (Thermo Fisher Scientific). Then, slides were incubated overnight at 4 °C with the primary antibodies. After rinsing with PBS, slides were incubated for 30 min at room temperature with the specific horseradish peroxidase (HRP)-conjugated secondary antibody (EnVision, Agilent Technologies). Slides were then developed using 3,3-diaminobenzedine tetrahydrochloride (DAB, Abcam), counterstained with Gill’s Hematoxylin N°2 (Sigma-Aldrich) and mounted using EuKitt (Bio-Optica). Micrographs were taken using Eclipse E800 microscope equipped with a cooled DS-U1 digital camera and NIS software (Nikon Instruments, Praha, Czech Republic; v. 5.30.00). The extent of the reactivity of the antibodies was semi-quantitatively scored as follows—0: absence of staining; 1: <5% of the cells (faint); 2: between 5 and 25% of the cells (focal); 3: between 25 and 75% of the cells (mild to diffuse); and 4: >75% of the cell (very diffuse).

### 4.7. Statistical Analysis

In vivo data values are expressed as means ± standard deviation (SD). The statistical significance of differences between treated and control groups was evaluated with unpaired Student’s *t*-test, whereas variations between pre- and post-treatment data per group were evaluated with paired Student’s *t*-test. The level of statistical significance was set at *p* < 0.05. Ex vivo data are generally given as mean values ± standard error of the mean (SEM), with n representing the number of experiments. Multiple comparisons were carried out with one-way ANOVA, followed by Tukey’s post hoc test, after checking for data normality (Kolmogorov–Smirnov test) and variance homogeneity (Brown–Forsythe test). The level of statistical significance was set at *p* < 0.05.

## Figures and Tables

**Figure 1 ijms-24-14124-f001:**
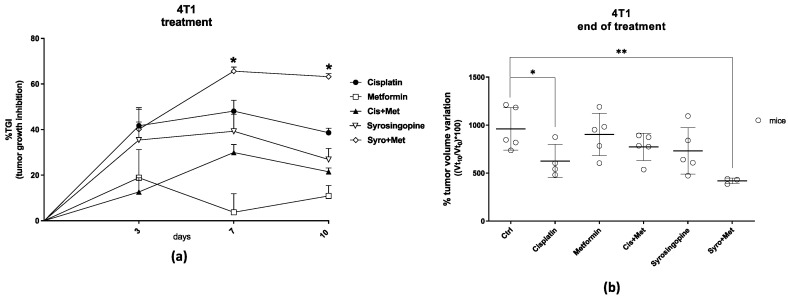
Treatment efficacy results on TNBC model (4T1 cell line), expressed as (**a**) percentage of tumor growth inhibition (%TGI) compared to the control group, calculated at different time points. Mice were randomly assigned to six groups of treatment: control (n. 5), cisplatin (n. 4, dose of 3 mg/kg i.p., twice a week), MET (n. 5, dose of 250 mg/kg i.p., daily), cisplatin plus MET (n. 5), SYRO (n. 5, dose of 7.5 mg/kg i.p., three times per week), and SYRO plus MET (n. 3). Average values calculated per group ± SEM; * (tumor growth inhibition > 52% is considered meaningful). Data at the end of treatment (10 days) were even expressed as (**b**) percentage of tumor volume variation compared to the start of treatment. Average values calculated per group ± SD (Student’s *t*-test; * *p* < 0.05 and ** *p* < 0.01 vs. vehicle group).

**Figure 2 ijms-24-14124-f002:**
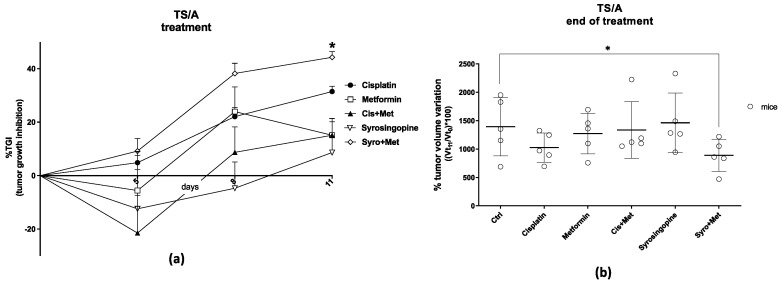
Treatment efficacy results on ER+ model (TS/A cell line), expressed as (**a**) percentage of tumor growth inhibition (%TGI) compared to the control group, calculated at different time points. Mice were randomly assigned to six groups of treatment: control (n. 5), cisplatin (n. 5, dose of 3 mg/kg i.p., twice a week), MET (n. 5, dose of 250 mg/kg i.p., daily), cisplatin plus MET (n. 5), SYRO (n. 5, dose of 7.5 mg/kg i.p., three times per week), and SYRO plus MET (n. 5). Average values calculated per group ± SEM; * (tumor growth inhibition > 39% is considered meaningful). Data at the end of treatment (11 days) were even expressed as (**b**) percentage of tumor volume variation compared to the start of treatment. Average values calculated per group ± SD (Student’s *t*-test; * *p* < 0.05 vs. vehicle group).

**Figure 3 ijms-24-14124-f003:**
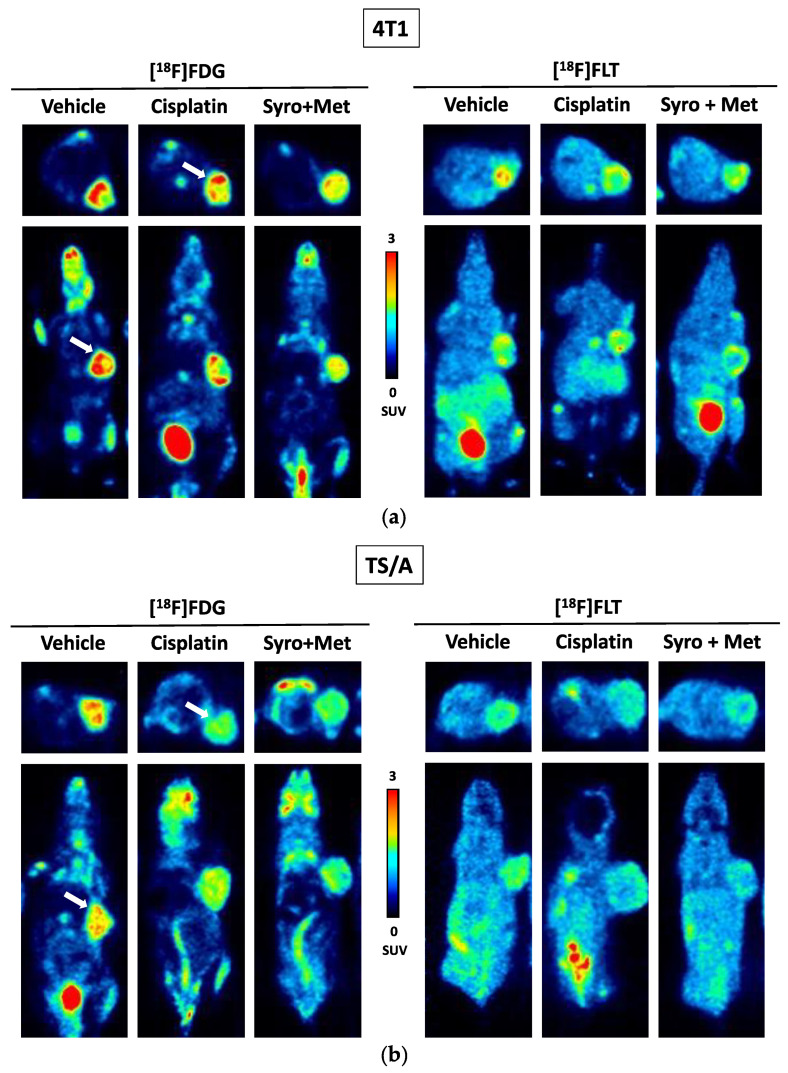
Representative PET images, performed at 7 days post treatment, of a (**a**) 4T1 and (**b**) TS/A xenograft mouse per group (vehicle (ctrl); cisplatin and syrosingopine plus metformin) injected i.v. with either [^18^F]FDG (**left**) or [^18^F]FLT (**right**) radiotracer (~4.7 MBq/mouse). Images are reported as standardized uptake mean value (SUV mean). White arrows indicate 4T1 or TS/A tumor.

**Figure 4 ijms-24-14124-f004:**
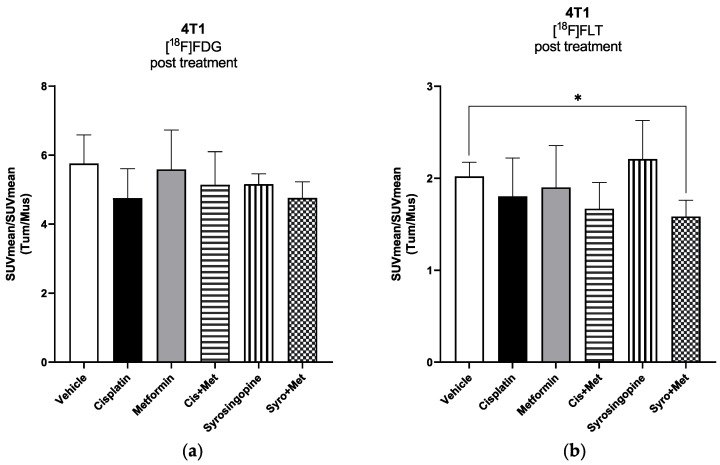
4T1 tumor-bearing balb/c mice (vehicle (ctrl): n. 5; cisplatin: n. 4; MET: n. 5; cisplatin plus MET: n. 5; SYRO: n. 5 and SYRO plus MET: n. 3) were injected i.v. with either (**a**) [^18^F]FDG or (**b**) [^18^F]FLT radiotracer (~4.7 MBq/mouse). Radiotracer uptake was assessed for each experimental group at 60 min post injection via whole-body PET/CT acquisitions performed post treatment. The quantification data are reported as tumor-to-muscle ratios of SUV mean values. Bars; mean ± SD (Student’s *t*-test; * *p* < 0.05 vs. vehicle group).

**Figure 5 ijms-24-14124-f005:**
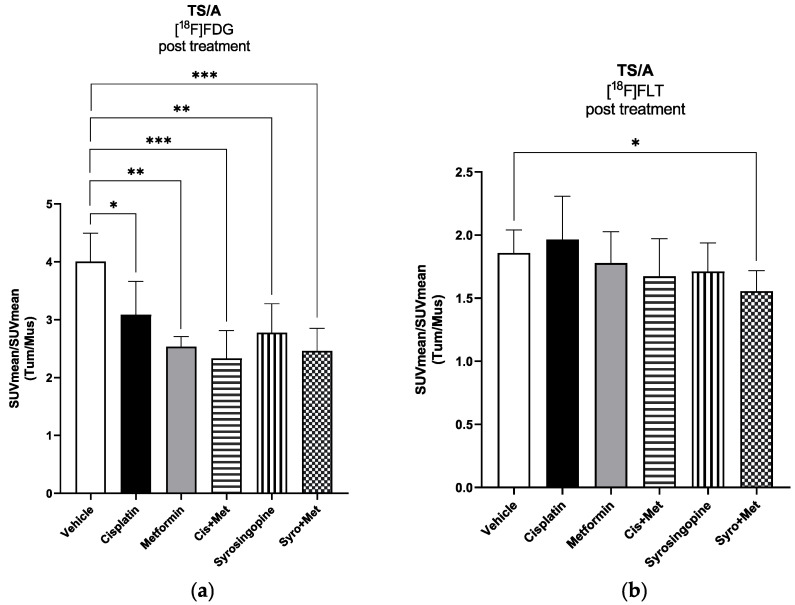
TS/A tumor-bearing balb/c mice (n. 5 mice per group) were injected i.v. with either (**a**) [^18^F]FDG or (**b**) [^18^F]FLT radiotracer (~4.7 MBq/mouse). Radiotracer uptake was assessed for each experimental group at 60 min post injection via whole-body PET/CT acquisitions performed post treatment. The quantification data are reported as tumor-to-muscle ratios of SUV mean values. Bars; mean ± SD (Student’s *t*-test; * *p* < 0.05, ** *p* < 0.01, and *** *p* < 0.001 vs. vehicle group).

**Figure 6 ijms-24-14124-f006:**
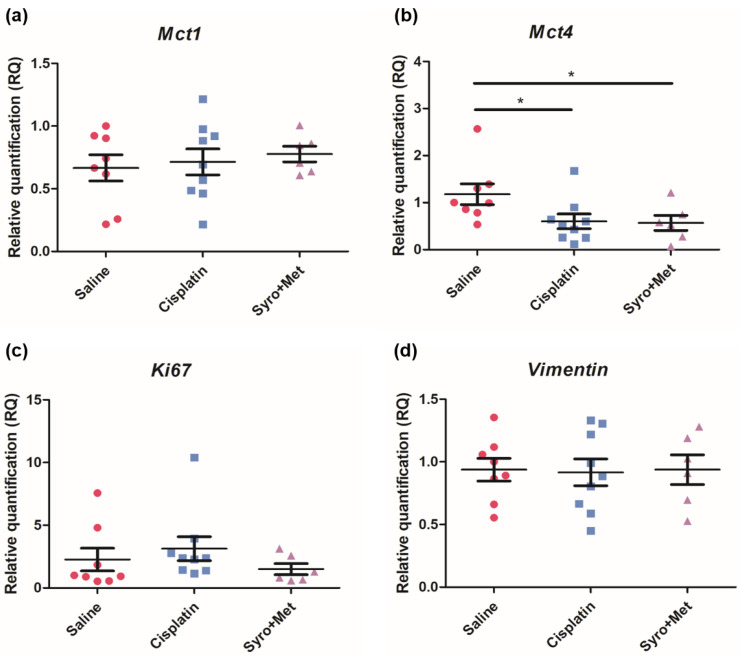
Expression levels of *MCT1* (**a**), *MCT4* (**b**), *Ki67* (**c**), *Vimentin* (**d**), *Snail* (**e**), *Slug* (**f**), *Mmp2* (**g**), and *Mmp9* (**h**) in tissue from 4T1 tumor-bearing balb/c mice (saline (ctrl): n. 8; cisplatin: n. 9, and syrosingopine plus metformin: n. 6). Data are expressed as fold increase ± SEM (Student’s *t*-test; * *p* < 0.05 vs. saline group).

**Figure 7 ijms-24-14124-f007:**
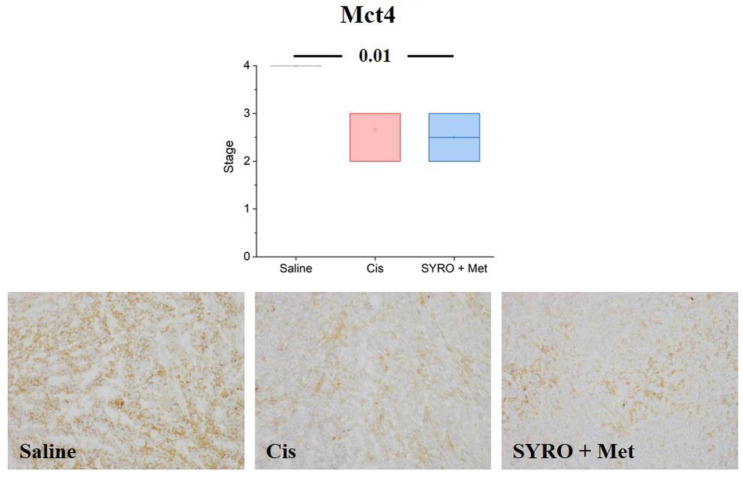
Immunohistochemical expression of Mct4 was semi-quantitatively scored, and results were showed using Box and Whiskers plot. Below-sided, representative micrographs showing the different extent of MCT4 expression in tumor samples were obtained from xenotransplanted mice challenged with different treatments: saline (ctrl), cisplatin (cis), and Syro plus met. Statistical analysis: one-way Analysis of Variance (ANOVA) test. Original magnification: 10×.

## Data Availability

Data are available in case of request.

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
