# Peer review of "Evaluating [18F]FDG and [18F]FLT Radiotracers as Biomarkers of Response for Combined Therapy Outcome in Triple-Negative and Estrogen-Receptor-Positive Breast Cancer Models"

_ijms, 2023, doi:10.3390/ijms241814124_

Round 1
Reviewer 1 Report
The authors present two radiotracers to evaluate the early response of TNBC cells to a challenge, in this case Metformin and Syrosingopine. Although they had previously done this with paclitaxel (Raccagni et al., PLoS ONE 13(5) 2018), they now reformulate it as an evaluation of early response. Therefore, the proposed work and the Plos ONE work share common points.
However, the use of [18F]FDG and [18F]FLT PET is not new. In 2008, Direcks et al. published an in vitro study in which they found that "cancer treatment can inhibit proliferation and may change FLT uptake, suggesting a role for FLT as a marker for monitoring response to chemotherapy" (British Journal of Cancer, 99, 481-487). The same idea was later published in 2021 as a measure of early response to treatment (Romine et al., Breast Cancer Res, 23:88).
In my opinion, the authors are not proposing anything new or original. This is the main weakness of this work for publication.
Reviewer 2 Report
In this study, the authors treated a triple negative murine breast cancer and a metastatic ER+ murine breast cancer with SYRO plus MET. FDG and FLT PET were used to evaluate the therapy response. Although statistic difference was found in the tumor volume and PET quantification, but the difference seems minor.
Major Comments:
1. In the treatment study, %TGI was used in Figure 1 and 3 without showing error bar. It’s difficult to compare the statistic difference. I would recommend to combine figure 1 to 4 to show the tumor volume curve. Besides, more animal number (at least 5) are needed to show a convincing treatment result.
2. PET imaging was performed at 7 d post treatment, when the reduction of tumor volume had been observed. It’s more like an evaluation of therapy response rather than the early prediction.
3. Introduction and Discussion should be separated into several paragraphs and reorganized to make it easier to understand.
Minor comments:
1. In the abstract, 4T1 is the triple negative murine BC cell line and TS/A is the metastatic ER+ murine BC cell line.
2. Representative PET images can be shown in Figure rather than supplementary date
Reviewer 3 Report
In this preclinical study the authors investigated the early response to treatment, based on MET and/or SYRO administration, evaluating the [18F]FDG and [18F]FLT as potential PET biomarkers of treatment outcome for triple negative and estrogen receptor positive breast cancer.
The study provides evidence that SYRO plus MET has a synergistic effect on tumor growth inhibition in preclinical models. Both [18F]FDG and [18F]FLT could be used as potential biomarkers of treatment outcome for triple negative and estrogen receptor positive breast cancer by PET/CT.
The article is clear, well-written and interesting. The methodology is adequate.
Minor suggestion: please add a comment in the discussion on the possible use of Fluoroestradiol as PET biomarker for the same indication (see PMID: 36879065 and PMID: 35407526)
Round 2
Reviewer 1 Report
I consider that the author has satisfactorily addressed my comments, so I believe the revised version of the manuscript is suitable for publication
Reviewer 2 Report
Most of my concerns were addressed in the author's response. Nevertheless, the limitation regarding the small number of animals studied remains a significant drawback in this paper. While the authors noted that efficacy was confirmed in the preliminary treatment study, the fact that the study was early halted prevents proper grouping and analysis. Additionally, it's noteworthy that the PET imaging data only involved 3 animals in the Syro+Met group, which consequently weakens the statistical validity of the comparisons made. I highly recommend considering a repetition of the study to increase the sample size.
Furthermore, I observed a disparity in the error bars within the TGI% figure. These error bars appear noticeably smaller, which contrasts with the results obtained from tumor volume measurements.
